# DNA Methylation Is Correlated with Gene Expression during Diapause Termination of Early Embryonic Development in the Silkworm (*Bombyx mori*)

**DOI:** 10.3390/ijms21020671

**Published:** 2020-01-20

**Authors:** Bing Li, Pei Hu, Lin-Bao Zhu, Ling-Ling You, Hui-Hua Cao, Jie Wang, Shang-Zhi Zhang, Ming-Hui Liu, Shahzad Toufeeq, Shou-Jun Huang, Jia-Ping Xu

**Affiliations:** 1School of Life Sciences, Anhui Agricultural University, Hefei 230036, Anhui, China; libing2504@sina.com (B.L.); m15755074091@163.com (P.H.); zhulinbao@163.com (L.-B.Z.); 13063306714@163.com (L.-L.Y.); chh18856960204@163.com (H.-H.C.); wangjie_3001@163.com (J.W.); 18755148780@163.com (S.-Z.Z.); toufeeq@163.com (S.T.); 2Institute of Sericulture, Anhui Academy of Agricultural Sciences, Hefei 230061, Anhui, China; mhliu-67@163.com; 3Anhui International Joint Research and Developmental Center of Sericulture Resources Utilization, Hefei 230036, Anhui, China

**Keywords:** DNA methylation, whole-genome sequencing, RNA-seq, embryonic development, diapause, *Bombyx mori*

## Abstract

DNA modification is a naturally occurring DNA modification in prokaryotic and eukaryotic organisms and is involved in several biological processes. Although genome-wide methylation has been studied in many insects, the understanding of global and genomic DNA methylation during insect early embryonic development, is lacking especially for insect diapause. In this study, we analyzed the relationship between DNA methylomes and transcriptomes in diapause-destined eggs compared to diapause-terminated eggs in the silkworm, *Bombyx mori (B. mori)*. The results revealed that methylation was sparse in this species, as previously reported. Moreover, methylation levels in diapause-terminated eggs (HCl-treated) were 0.05% higher than in non-treated eggs, mainly due to the contribution of CG methylation sites. Methylation tends to occur in the coding sequences and promoter regions, especially at transcription initiation sites and short interspersed elements. Additionally, 364 methylome- and transcriptome-associated genes were identified, which showed significant differences in methylation and expression levels in diapause-destined eggs when compared with diapause-terminated eggs, and 74% of methylome and transcriptome associated genes showed both hypermethylation and elevated expression. Most importantly, Kyoto Encyclopaedia of Genes and Genomes (KEGG) analyses showed that methylation may be positively associated with *Bombyx mori* embryonic development, by regulating cell differentiation, metabolism, apoptosis pathways and phosphorylation. Through analyzing the G2/M phase-specific E3 ubiquitin-protein ligase (*G2E3*), we speculate that methylation may affect embryo diapause by regulating the cell cycle in *Bombyx mori*. These findings will help unravel potential linkages between DNA methylation and gene expression during early insect embryonic development and insect diapause.

## 1. Introduction

DNA methylation is an important epigenetic modification in plants and animals [1,2,3,4]. It has been shown to play key roles in a broad range of biological processes, including embryonic development [5], histone modification [6], X chromosome inactivation [7] and brain development [8]. In mammals, a primary function of DNA methylation is to suppress gene expression through increased promoter DNA methylation [9]. The roles of methylation include regulation of transcriptional activities [1], alternative exon splicing [10] and developmental activities [11]. DNA methylation in animals is accomplished by two types of DNA methyltransferases (DNMTs), de novo and maintenance DNMTs [12]. De novo DNMTs are responsible for establishing new methylation patterns by the Dnmt3 family of proteins, and maintenance DNMTs maintain previously established methylation patterns and are represented by the Dnmt1 family of proteins [13]. In contrast to these DNMTs, Dnmt2 shows a weak methyltransferase activity towards DNA and has been characterized as an active RNA methyltransferase [12,14]. Although *Drosophila melanogaster* possesses only a single DNMT (Dnmt2), the DNA methylation rate in the genome is at a low detectable level [15].

In insects, DNA methylation is found at appreciable levels in the genomes of many, but not all, insects [16] and is primarily targeted to genes that are broadly expressed in the genome [17]. The presence of a functional DNA methylation system across the insect class with conserved patterns of methylation [18,19,20], suggests an important role for this epigenetic marker in insect biology. The importance of DNA methylation in insect development has been strikingly demonstrated in *Apis mellifera*, which is capable of influencing developmental plasticity [21,22,23]. Chen et al. provided evidence for a functional DNA methylation system in *Plutellaxylostella* and its possible role in adaptation during host transfer [24]. However, fundamental questions remain on the patterning of DNA methylation during insect development; in particular, how DNA methylation affects the functional mechanism of early insect embryonic development and insect diapause.

Advances in whole-genome sequencing, coupled with bisulfite DNA treatment (WGBS) have led to single-nucleotide resolution methylation maps in a wide range of invertebrates [10,25,26]. Most recently, Jones et al. used WGBS and behavioural assays in a detailed analysis of the *Helicoverpaarmigera* methylome to investigate potential methylation differences in a subset of genes, demonstrating distinct flight performances in insects [27]. Similarly, Guan et al. examined the effects of Cd exposure on global DNA methylation using WGBS in *Drosophila melanogaster* [15]. In the silkworm, WGBS was used to investigate the silkworm mid-gut, identifying 0.11% methylcytosine levels across genomic cytosines [28]. Recently, Wu et al. used WGBS to demonstrate that epigenetic regulation may play roles in host–virus interactions in the silkworm [29]. As a robust and effective technique, WGBS is becoming more and more popular for insect DNA methylation research, including in the silkworm.

The silkworm is an economically important model insect of the lepidopteran order. Previous silkworm studies have shown that ~0.11% of their DNA is methylated [28], with only *Bombyx moriDnmt1* (*BmDnmt1*) and *Bombyx moriDnmt2* (*BmDnmt2*) proteins being reported [28,30], and *BmDnmt1* retains the function as maintenance DNMT, but its sensitivity to metal ions is different from mammalian Dnmt1 [31]. Silkworm, like *Polistes canadensis* [32], has lost orthologs of Dnmt3 [28], and BmDnmt1 preferentially methylates hemimethylated DNA [31]. Kay et al. suggests that *Dnmt1* functions primarily as a maintenance DNMT in *B. mori*, despite a putative secondary role in de novo methylation [33]. By contrast, Dnmt2 has been characterized as an active RNA methyltransferase [14], and the function of BmDnmt2 is unclear. Silkworms can revert to a diapause state, which arrests development in early embryonic development stages. The state is used to survive unfavourable environments such as low temperatures, drought and/or food shortages [34,35]. Diapause occurs at specific embryonic stages, i.e., after formation of cephalic lobes and telson and sequential mesoderm segmentation [36,37]. Diapause-destined eggs completely enter diapause three days after oviposition when non-HCl treated. If diapause-destined eggs are treated with 1.075 g/L HCl 24 h after oviposition, the eggs will terminate diapause 48 h after HCl treatment [34]. The development of diapause-destined embryos is arrested during the G2 cell cycle stage, immediately after mesoderm formation [38]. During sericulture, hydrochloric acid (HCl) is often used to treat silkworm eggs to disrupt diapause [39]. Once diapause terminates, the embryos resume development at 25 °C, they enter M phase, and cell division resumes [38]. The silkworm is an ideal model for studying relationships between DNA methylation, embryonic development and insect diapause.

Currently, functional analyses of DNA methylation during insect embryogenesis, especially insect diapause, are limited. Our previous study provided functional insights into *BmDnmt1* and *BmDnmt2* in the regulation of silkworm embryonic development; we showed that the expression of *BmDnmt1* and *BmDnmt2* was elevated in diapause-terminated eggs that were HCl-treated when compared with diapause-destined eggs [40].

In this study, we investigate DNA methylation patterns using WGBS and RNA sequencing (RNA-seq) technologies in diapause-terminated eggs compared with diapause-destined eggs in *B. mori*. We explore different methylation patterns and methylation modification genes and pathways associated with embryonic development. Based on our findings, DNA methylation appears to be essential during *B. mori* diapause and embryogenesis.

## 2. Results

### 2.1. Transcriptional Profiles of Silkworm Diapause-Destined Eggs and Diapause-Terminated Eggs

To explore gene expression mechanisms during early embryonic development in the silkworm, we compared the transcriptional profiles of diapause-terminated eggs treated with HCl (HCl-treated) and diapause-destined eggs (non-HCl-treated) using RNA-seq, and the raw sequencing data are available from the repository of NCBI Sequence Read Archive (SRA) with the accession No. PRJNA598980. More than 48 million clean reads were obtained from each sample. The CG content of each of the six libraries was approximately 44%, and CycleQ30 was >94% for each library. The proportion of total reads that mapped to the reference genome reached approximately 95% coverage. In addition, 1732 differentially expressed genes (DEGs) were identified, with 856 genes upregulated and 876 genes downregulated (Table 1). The details of DEGs are shown in Appendix A. Gene Ontology (GO) analysis showed that common DEGs were mainly distributed to binding, transporter activity, locomotion, catalytic activity (Figure 1A). Kyoto Encyclopaedia of Genes and Genomes (KEGG) pathway analyses showed that DEGs were involved in cellular processes, environmental information processing, genetic information processing and metabolism and organismal systems pathways (Figure 1B). It is noteworthy that the most upregulated genes were enriched for signal transduction and carbohydrate metabolism pathways, with 39 and 30 DEGs, respectively. In addition, the most downregulated genes were enriched for lipid metabolism and amino acid metabolism pathways, with 24 and 19 DEGs, respectively.

The expression of *BmDnmt1* (and *BmDnmt2* to a lesser extent) was upregulated in diapause-terminated eggs after HCl-treatment. This was consistent with our previous findings [40] and consistent with RNA-seq data generated in this study (Figure 2). These changes in DNA methyltransferase expression suggest that genomic DNA methylation patterns may be associated with diapause initiation and termination in silkworm embryonic development.

### 2.2. Overview of Methylation Landscapes of Silkworm Diapause-Destined Eggs and Diapause-Terminated Eggs

To investigate methylation patterns in silkworm embryonic development, WGBS was conducted in diapause termination eggs treated with HCl (HCl-treated) and diapause-destined eggs (non-HCl-treated). Three repeats were performed for each of the two groups, and the raw sequencing data are available from the repository of NCBI Sequence Read Archive (SRA) with the accession No. PRJNA598995. Up to ~9000 million sequencing reads were generated for each replicate, yielding ~12 Gb of data representing >30× the NCBI reference genome for silkworm. Nearly 98% of the reads (bases) were retained and were high-quality clean reads (bases) (Table 2). The non-HCl sample (control) genome presented ~0.21% mC on the total sequenced C sites, which reflected the methylation level percentage in the genome. It contained ~98.55% (mCG), 0.27% (mCHG) and 1.18% (mCHH) on the total sequenced mC sites. Accordingly, the HCl-treated sample presented ~0.26% of the total sequenced C sites, and 98.94%, 0.21% and 0.85% of the CG, CHG and CHH sites, respectively (Appendix A). It was observed that the mC rate of the total sequenced C sites was significantly increased in diapause-terminated eggs when compared with diapause-destined eggs (Figure 3A). In addition, our results showed that mCG sites were significantly increased in diapause-terminated eggs when compared with diapause-destined eggs, while others were not significantly changed (Figure 3B).

### 2.3. Validation of RNA-seq and WGBS

For the validation of RNA-seq results, three genes that were significantly upregulated and three genes that were significantly downregulated were selected for mRNA quantification using RT-qPCR. The results were consistent with RNA-seq (Figure 4A). McrBC is a restriction enzyme that cuts methylated, but not unmethylated, DNA. McrBC-qPCR has been routinely used for DNA methylation testing and marker validation [41,42]. For WGBS validation, we assayed DNA methylation levels of two genomic regions using methylation-sensitive McrBC-qPCR. The gene LOC101739208 is hypermethylated, and we determined that its methylation levels increased by 28% in HCl-treated samples when compared with non-treated samples. In another experiment using the gene LOC110385781, which is hypomethylated, we observed that methylation levels decreased by 25% in HCl-treated samples when compared with non-treated samples. Methylated DNA can be digested by McrBC; thus, higher qPCR signals indicate lower methylation levels, suggesting our McrBC-qPCR assay was consistent with our WGBS data (Figure 4B).

### 2.4. DNA Methylation Patterns in Genomic Regions of Diapause-Destined Eggs and Diapause-Terminated Eggs

From our data, we observed that DNA methylation patterns varied at different genomic intervals. To understand these patterns in silkworms, we analyzed methylation profiles within genes, including Coding sequence (CDS), downstream, exons, intergenic areas, introns, lnc-RNAs, miRNAs, mRNAs, transcripts and upstream (Figure 5A). When compared with the non-treated group, CG methylation levels increased in the HCl-treated group for all types of genomic intervals, with the highest methylation levels found in transcript elements. Methylation profiles were also analyzed in transposon elements (TEs) and repeat elements (REs) using RepeatMasker software (GIRI, v4.0.4). Transposons include DNA segments, long interspersed nuclear elements (LINE), short interspersed nuclear elements (SINE), retrotransposons (RC) and long terminal repeats (LTRs), and repeat elements include simple_repeat, low_complexity, satellites and RNArepeats (Figure 5B); in particular, satellites of HCl treated were not detected. The results suggest that CG context methylation sites are mainly concentrated in SINE repeats, DNA transposons and RC transposons.

To understand the relationship between DNA methylation profiles and gene expression, DNA methylation profiles were divided into ~2 kb upstream and downstream transcribed regions to study methylation changes (Figure 6). The distribution of CG methylation sites in three distinct genomic functional regions showed strong bias, and CG methylation showed high enrichment levels in ~2 kb upstream, transcription initiation sites (TSS) downstream and transcription termination sites (TTS) downstream regions. In addition, CG methylation levels in HCl-treated samples were elevated among gene features when compared with non-treated samples, except for regions near TSS and TTS.

### 2.5. Differentially Methylated Regions (DMRs) and Differentially Methylated Promoters (DMPs) Responding to Embryonic Development

To investigate the influence of embryonic development and insect diapause on methylation, we analyzed DMRs in diapause-terminated eggs when compared with diapause-destined eggs. In total, 43,781 DMRs were identified (methylation difference ≥ 10%, Q value ≤ 0.05) containing 42,877 hypermethylated DMRs and 904 hypomethylated DMRs. In addition, 4449 different methylation genes (DMGs) located in DMRs were identified. To investigate the biological functions of DMGs, a KEGG pathway enrichment analysis was performed. As shown in Figure 7A, DMGs were mainly assigned to pathways related to protein processing in endoplasmic, cell cycle, RNA transport, etc. functions.

The promoter region of a gene is closely related to transcriptional regulation. Differentially methylated modifications located in the promoter region may change a gene’s transcriptional expression. To characterize such methylation alterations in promoter regions during early embryonic development, 99 differentially methylated promoters (DMPs) located in DMRs were identified (Appendix A). KEGG pathway analyses showed that DMPs were involved in hippo-signalling, Wnt-signalling, mTOR-signalling, etc. pathways. (Figure 7B).

### 2.6. Correlations Between DNA Methylation and Gene Expression during Silkworm Early Embryonic Development

To investigate whether changes in CG methylation observed during silkworm early embryonic development altered gene expression, we synthetically analyzed DEG and DMG data. As a result, there were 364 methylome- and transcriptome-associated genes (MTGs) with significant differences in methylation and expression levels in diapause-terminated eggs when compared with diapause-destined eggs (Appendix A). More than 98% of genes showed hypermethylation, and more than 74% of genes showed both hypermethylation and upregulated protein expression. To be specific, 270 hypermethylated genes were positively correlated with expression changes in the first quartile, and 88 hypermethylated genes were negatively correlated with expression changes in the second quartile (Figure 8A), indicating a strong positive correlation between DNA methylation and gene expression for first quadrant genes. Meanwhile, percentage methylation levels in gene body, 2 kb upstream and downstream regions for positively and negatively correlated genes, were detected, to understand the relationship between DNA methylation profiles and gene expression (Appendix A). The results showed that CG methylation sites of 270 hypermethylated-upregulated genes showed strong bias. The CG methylation levels in HCl-treated samples were strongly elevated in 2 kb downstream and weakly elevated near the TSS of gene body when compared with non-treated samples. By contrast, there is no obvious change trend for the 88 hypermethylated-downregulated genes. Moreover, GO analysis showed that MTGs were highly enriched for phosphatidylinositol dephosphorylation, G1/S transition in the mitotic cell cycle, RNA polymerase II transcription corepressor activities etc., (Figure 8B), and related genes with changes in methylation levels or methylation patterns may be closely correlated with embryonic development. KEGG pathway analyses showed that MTGs were involved in signalling pathways regulating stem cell pluripotency, phosphatidylinositol signalling system, different types of N−glycan biosynthesis, insulin-signalling pathways etc. (Figure 8C), suggesting that DNA methylation variations appear to regulate gene expression in multiple pathways during early embryonic development.

### 2.7. MTGs Are Involved in Metabolic Biosynthesis

In embryonic development, when the diapause eggs have terminated the diapause, energy metabolism and material synthesis begin to be activated. Here, KEGG pathway analyses showed that 23 MTGs related to metabolic biosynthesis were identified for pathways involved in N-glycan biosynthesis, glycosaminoglycan biosynthesis, glutathione metabolism, regulation of lipolysis in adipocytes, biosynthesis of amino acids and glyoxylate and dicarboxylate metabolism (Table 3). These pathways were associated with metabolism and synthesis of glycogenolysis, protein synthesis and lipolysis. Moreover, the most gene-enriched pathway was the protein-synthesis-related pathway, in which 16 MTGs were identified. Previous studies have shown that glycogen synthesis and tricarboxylic acid cycle dynamics are enhanced during embryonic development [43]. Here, four MTGs were involved in glycogen synthesis and three MTGs in the tricarboxylic acid cycle. Three MTGs involved in lipid metabolism were also identified; it is worth noting that a homologue of adenylate cyclase type 2 (Gene ID: LOC101737606) had a sevenfold increase in protein expression. Thus, KEGG analyses demonstrated that methylation modification was related to embryonic metabolic biosynthesis.

### 2.8. MTGs Are Involved in Phosphorylation

A previous study has shown that some key enzymes in silkworm embryo development regulate their activities through protein phosphorylation and dephosphorylation to sustain embryonic metabolism [44]. Our KEGG pathway analyses showed that six MTGs were involved in phosphatidylinositol-signalling pathways (Table 4), with all genes showing increased methylation and expression levels in diapause-terminated eggs when compared with diapause-destined eggs. It is noteworthy that three INPP-family genes of the six MTGs were identified, and that INPP genes regulate many cellular activities, including vesicle transport, cytoskeletal dynamics, protein synthesis, cell proliferation and survival [45]. Based on these observations, our data indicate that methylation may play a role in embryonic development by regulating phosphorylation.

### 2.9. MTGs are Involved in the Cell Cycle, Apoptosis and Stem Cell Pluripotency

Silkworm embryonic cells are arrested in G2 at diapause, and cell cycles become slower in proportion to an increasing length of G1. Once diapause terminates, the embryos are competent to resume development at 25 °C, cells enter the M phase, and cell division then resumes in the embryos [38]. Based on this evidence, we hypothesized that embryonic development and diapause were related to cell cycle regulation. Here, KEGG pathway analyses showed that 16 MTGs were involved in G1/S transition of the mitotic cell cycle, regulation of apoptotic processes and signalling pathways regulating stem cell pluripotency (Table 5). Moreover, six MTGs were enriched in the G1/S transition of the mitotic cell cycle, and we observed that 101739208, a homologue of the G2/M phase-specific E3 ubiquitin-protein ligase (*G2E3*), was upregulated and hypermethylated. Alternately, diapause-destined embryos are arrested at gastrulation, which occurs two days after the germ-band formation stage (24 h after oviposition). At this stage, cell differentiation gradually ceases, and the embryo gradually forms [46]. KEGG analyses revealed that six MTGs were enriched in signalling pathways regulating stem cell pluripotency, which increased gene expression. Those MTGs can promote cell differentiation and organ formation in diapause-terminated eggs. Based on this evidence, these data suggest that diapause-destined eggs may regulate the cell cycle, apoptosis and stem cell pluripotency to terminate diapause via methylation.

## 3. Discussion

DNA methylation is one of the most widespread epigenetic markers in the genome and has been linked to insect development, specifically embryonic development [16,47]. Our previous study showed that *BmDNMT*s were highly expressed during embryonic development, especially at early embryonic stages; its expression was significantly upregulated in diapause-terminated eggs after HCl-treatment [40]. However, DNA methylation regulation in silkworm embryonic development and diapause still remains unclear. This study provided insights into the potential cytosine methylation in diapause-destined eggs and diapause-terminated eggs using WGBS in relation to methylation and embryonic development.

Overall, the levels of DNA methylation in most insects is <1% [28,48,49]. This research has shown ~0.21%–0.26% mC methylation levels on the total sequenced C sites, and more than ~98.5% mC methylation in CG sequences. Based on previous methylome studies in the silkworm, the average methylation levels in the mid-gut and fat body at mC methylation in the genome were approximately 1% [29], and ~0.11% for silk glands [28]. When comparing eggs, the mid-gut, fat body and silk gland, it has been shown that DNA methylation levels have tissue specificity in the silkworm. Moreover, mC rates in sequenced C sites were significantly increased in diapause-terminated eggs (HCl-treated) when compared with diapause-destined eggs (non-treated), and change in methylation levels mainly comes from CG sites but not from CHG and CHH sites. It is suggested that when compared with other plants and animals which have higher methylation levels at CHG and CHH sites [50,51], DNA methylation works primarily through CGs site in the silkworm.

DNA methylation generally functions as a repressive transcriptional signal, but it is also known to activate gene expression [52,53]. Here, we calculated methylation levels in the context of gene regions and ~2 kb upstream and downstream regions. Consistently, CG sites showed more regularity than CHG and CHH sites. CG methylation, but not CHG or CHH methylation, exhibited a characteristic peak in the body of protein-coding genes and showed significant increases after TSS and TES, which maintain low methylation levels. These results are similar to those previously reported for other species [54]. Moreover, CG methylation levels in the HCl-treated group were generally higher than those in the non-treated group. Thus, we speculate that CG methylation occurs in gene body regions, enhancing gene transcription for embryonic development.

To investigate the specific modification regions of DNA methyl groups in the silkworm genome, methylation profiles of genetic elements were analyzed, including CDS, downstream, upstream, exons, etc. Notably, methylation levels in all genetic elements were increased in HCl-treated samples when compared with non-treated samples, and methylation mainly occurred in coding sequences such as transcript, CDS and exons. Previous studies in silkworms have shown that CG methylation is substantially enriched in gene bodies and is positively correlated with gene expression levels, suggesting positive roles in gene transcription [26,28]. Therefore, hypermethylation of coding sequences may be positively associated with the terminating of diapause in silkworm embryos.

DNA methylation patterns are prevalent in TEs and REs [17]. Our results show that methylation in TEs, especially SINEs, DNA transposable, and RC are higher than in others. SINEs are an abundant class of TEs found in a wide variety of eukaryotes, mainly integrate into hypomethylated DNA regions and are targeted by methylases for de novo methylation [55,56,57]. Previous studies have shown that SINEs promote the amplification of gene enrichment region fragments or stress-induced gene expression, which increase gene expression [58]. Based on these analyses, hypermethylated SINEs may promote gene transcription and expression during embryo development.

Differentiation and organogenesis are two key phases of embryogenesis in the silkworm. Metabolic biosynthesis is also an essential activity during this period. Metabolic pathways were significantly activated after HCl-treatment, especially for glycogenolysis, protein synthesis and lipolysis. It is worth noting that *HS6ST2* homologous gene (GeneID: LOC101737390) regulates energy metabolism and promoted embryonic development in *Drosophila* [59,60]. Moreover, the *Pi3k60* homologous gene (GeneID: LOC100158253), which was upregulated and hypermethylated, reportedly promotes the decomposition of lipids into energy for embryonic development [61,62]. Thus, these results suggest that methylation modification may influence embryo development by regulating metabolic biosynthesis phases.

Enzyme inactivation via protein phosphorylation regulates embryo development during embryogenesis [44]. Here, six MTGs were identified as being involved in phosphatidylinositol signalling. These genes showed increased methylation and expression level in diapause-terminated eggs when compared with diapause-destined eggs. Ponnuvel et al. reported that enzyme inactivation via protein phosphorylation during early silkworm embryogenesis was followed by dephosphorylation in later stages [44]. On the other hand, INPP genes are important in regulating many cellular activities, including vesicle transport, cytoskeletal dynamics, protein synthesis, cell proliferation and survival and phosphatidylinositol signalling [45]. Here, *Inpp5j*, *Inpp5e* and *Inpp3a*, three of the six MTGs involved in the phosphatidylinositol dephosphorylation pathway, showed increased expression and methylation levels. It was also reported that Inpp5e promotes fundamental metabolism in fat body via phosphorylation [63]. These results indicate that phosphorylation pathways are activated after gene methylation, thereby promoting embryonic signal transduction, material synthesis and other key activities during embryo development.

Our previous data showed that embryonic cells are arrested in G2 at diapause, and cell cycles become slower in proportion to increasing G1 length. Once diapause terminates, the embryos resume development at 25 °C, cells enter M phase, and cell division resumes in the embryos [38]. GO analysis revealed that six MTGs were enriched in the G1/S transition of the mitotic cell cycle. It was interesting that gene 101739208, one of the six MTGs, was a homologue of the G2/M phase-specific E3 ubiquitin-protein ligase (*G2E3*) and was upregulated and hypermethylated by HCl-treatment. BS-PCR results showed that non-treated and HCl-treated sample methylation levels were 33.3% and 48.1%, respectively for *G2E3* (Figure 9), consistent with WGBS data. Previous research has shown that *G2E3* is essential for early embryonic development in preventing apoptotic death, and it strengthens the synthesis and transport of genetic material and energy in the M phase [64]. This indicates that hypermethylation is associated with embryonic diapause and may be regulated through the cell cycle and apoptosis inhibition.

## 4. Materials and Methods

### 4.1. Silkworm Feeding and Artificial Diapause Termination

Bivoltine silkworm strain P50 was maintained in the School of Life Sciences, Anhui Agricultural University, Hefei, China. Instar larvae were reared on fresh mulberry leaves at 25 ± 1 °C, 75 ± 5 % relative humidity, with a 12 h day/night cycle. Female moths were mated with males for more than five hours and then separated to allow the females to lay eggs. For artificial diapause termination, diapause-destined eggs were treated with 1.075 g/L HCl at 46 °C for 5 min, 24 h after oviposition. Eggs for incubation were maintained at 25 °C, and a 12 h light/12 h dark period was observed.

### 4.2. Sample Preparation for RNA-seq and WGBS

Diapause-destined eggs were collected 3 days after oviposition (non-HCl treated control group). Diapause-terminated eggs were treated with 1.075 g/L HCl at 46 °C for 5 min, 24 h after oviposition, which terminated the diapause at 48 h after HCl treatment (3 days after oviposition, experimental group). Each sample was taken from five individuals, and three duplicate samples were taken after mixing. Both experimental and control group eggs were collected on Day 3 following oviposition and quickly frozen in liquid nitrogen and subsequently powdered. Half of the samples from each powdered group were treated with Trizol reagent (Invitrogen, Carlsbad, CA, USA) to extract RNA for RNA-seq assay, and DNA was isolated from the other half using the DNeasy Blood and Tissue Kit (Qiagen, Dusseldorf, Germany) for WGBS assay.

### 4.3. WGBS

For WGBS, analyses were conducted by OE Biotech Co. Ltd. (Shanghai, China). DNA was fragmented by sonication in a Bioruptor (Diagenode, Brussels, Belgium) to a mean approximate size of 250 bp. This was followed by adding dA to the 3′ end by blunt-end cloning and methylated adaptor ligation. Ligated DNA was bisulfite-converted using the EZ DNA Methylation-Gold kit (ZYMO, Tustin, CA, USA). Different DNA fragment lengths were excised from a 2% agarose gel and purified and amplified by PCR. Finally, sequencing was performed using the Illumina Hiseq 4000 platform (Illumina, San Diego, CA, USA). After filtering, the clean data were mapped to the silkworm genome (*B. mori* assembly ASM15162v1) using the whole-genome bisulfite sequencing mapping program (BSMAP, v2.74) [65]. Then, mapping rates and bisulfite conversion rates were calculated for each sample.

Differentially methylated regions (DMRs) were identified by swDMR online software (http://122.228.158.106/swDMR/), which used a sliding-window approach. The window was 1000 bp, and the step length was 100 bp. The Fisher test was applied to detect significant DMRs, and the screened criteria as methylation differences ≥10%, Q value ≤ 0.05. After DMRs were identified, differentially methylated genes (DMGs) located in DMRs were characterized.

### 4.4. Transcriptome Analysis

Transcriptome sequencing and analyses were conducted by OE Biotech Co. Ltd. (Shanghai, China). Briefly, mRNA was enriched using oligonucleotide (dT) magnetic beads and fragmented at an elevated temperature. Double-stranded cDNA was synthesised and purified for each targeted fragment (200–500 bp). Libraries were constructed using TruSeq Stranded mRNA LTSample Prep Kit (Illumina, San Diego, CA, USA) according to manufacturer’s instructions. These libraries were then sequenced on the Illumina HiSeq X Ten sequencing platform, generating 125 bp/150 bp paired-end reads. Raw data (raw reads) were processed using Trimmomatic [66]. Reads containing poly-N tails and low-quality reads were removed to derive clean reads. These were then mapped to silkworm mRNA (*B. mori* assembly ASM15162v1) using HISAT2 (version 2.2.5, Johns Hopkins University, Washington, DC, USA) [67]. Differentially expressed genes (DEGs) were identified by EBseq (version 1.4.0, Bioconductor, Madison, Wisconsin, USA), with screened criteria as fold change ≥ 2 and *p*-values ≤ 0.05.

### 4.5. GO and KEGG Enrichment Analysis

Gene Ontology (GO) enrichment analyses of DMGs and DEGs were implemented by the GO-seq R package [68], where gene length biases was corrected. GO terms with corrected *p*-values < 0.05 were considered significantly enriched by DMGs and DEGs. The Kyoto Encyclopaedia of Genes and Genomes (KEGG: http://www.genome.jp/kegg/) [69] is a database that provides high-level function annotation for a diverse range of biological systems. The KOBAS software (version 3.0, Chinese Academy of Sciences, Beijing, China) was used to test for statistical enrichment of DMGs and DEGs in KEGG pathways [70]. KEGG pathway annotation and enrichment analyses were performed similar to GO analysis.

### 4.6. RT-qPCR and McrBC-qPCR Analysis

Total RNA for RNA-seq analyses was also used for RT-qPCR. The primers for RT-qPCR are shown (Appendix A). RT-qPCR reactions were performed in 25 μL reaction volumes containing SYBR Premix Ex Taq (TaKaRa, Dalian, China) according to manufacturer’s instructions. The reaction was performed in a CFX96TM Real-Time System (Bio-Rad, Hercules, CA, USA). Thermal cycling profiles consisted of an initial denaturation at 95 °C for the 30s and 40 cycles at 95 °C for 5 s and 60 °C for 30 s. GO and KEGG pathway analysis identified some important pathways that may be related to diapause; we then selected genes from these pathways for verification from the list of DEGs presented in Appendix A, based on their functions: *CG14226* (101742044), *CG7766* (101739131), *DDX5* (101736027), *ZFP39* (110385781), *MBD4* (101740063), *HIGD1A* (101741941), *G2E3* (101739208). For McrBC-qPCR, 1 µg of genomic DNA for WGBS was digested overnight with McrBC (New England Biolabs, MA, USA). The negative control was performed without GTP for digestion of the standard. qPCR was performed using 20 ng DNA as template. The qPCR process was the same as described earlier, and primers used are shown (Appendix A). Three independent experiments, with three technical replicates, were performed. All assays were performed in triplicate, and relative expression levels were calculated using the 2^−ΔΔCt^ method [71]. Statistical analyses were conducted using analysis of variance (ANOVA) and a least significant difference (LSD) posteriori test using SPSS software (version 19.0, IBM, Chicago, USA).

### 4.7. Bisulfite Sequencing Validation of Different Methylation Genes

Genomic DNA was bisulfite-converted using the EZ DNA Methylation-Gold kit (ZYMO, Tustin, CA, USA) according to manufacturer’s instructions. Bisulfite-converted DNA was amplified by PCR using Premix Ex Taq™ Hot Start Version (TaKaRa, Dalian, China) according to manufacturer’s instructions. Primers for BS-PCR were designed using the online MethPrimer program (http://www.urogene.org/cgibin/methprimer2/Meth Primer.cgi) (Appendix A). PCR products were purified and cloned into the pMD19-T vector (TaKaRa, Dalian, China), and six clones, selected from each sample, were sent to Sangon Biotech Co. Ltd. (Shanghai, China) for sequencing. The sequencing results were analyzed using the online Quma software (http://quma.cdb.riken.jp/) for individual CG sites and DNA methylation levels.

## 5. Conclusions

In conclusion, epigenetic modification is a significant dimension to embryonic development in insects. Our findings revealed that methylation levels in diapause-terminated eggs were higher than in diapause-destined eggs in silkworm and that methylation sites in diapause-terminated eggs tended to be in coding sequences and promoter regions when compared with diapause-destined eggs. Most importantly, our result indicated that methylation was positively associated with embryonic development in silkworm, by regulating cell differentiation, cell cycle, metabolism, apoptosis and phosphorylation. These data will be useful in understanding molecular mechanisms underlying DNA methylation and gene expression changes during embryonic development and insect diapause.

## Figures and Tables

**Figure 1 ijms-21-00671-f001:**
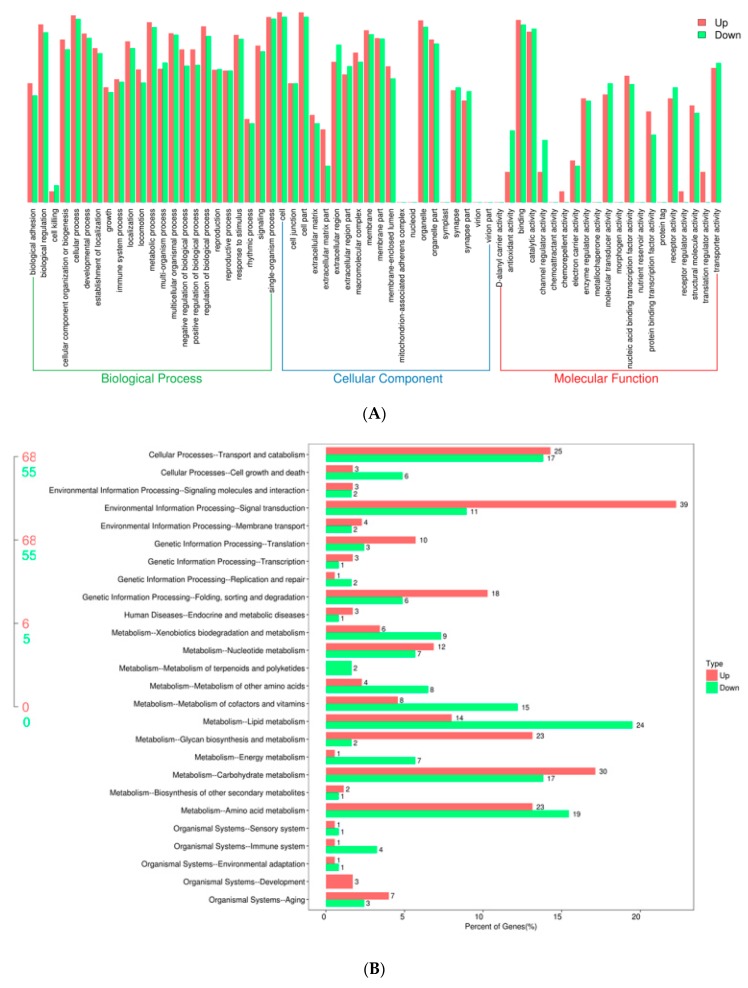
Enrichment analysis of differentially expressed genes (DEGs) following HCl treatment. (**A**) Annotation of DEGs with GO enrichment. Gene numbers and percentages are listed for each category. (**B**) Kyoto Encyclopaedia of Genes and Genomes (KEGG) pathway enrichment of DEGs. The red and green bars represent upregulated and downregulated genes, respectively, and the number represents the percentage of genes.

**Figure 2 ijms-21-00671-f002:**
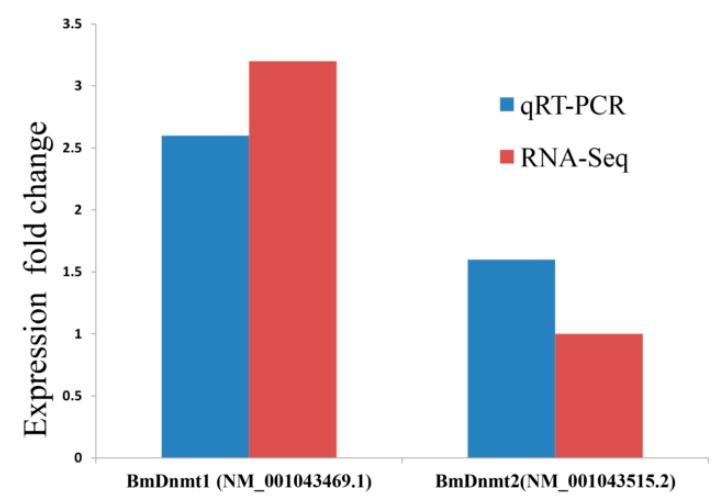
Analysis of methyltransferase expression. The expression fold change of *BmDnmt1* and *BmDnmt2* by qRT-PCR and RNA-seq, respectively, comparing diapause-terminated eggs with diapause-destined eggs.

**Figure 3 ijms-21-00671-f003:**
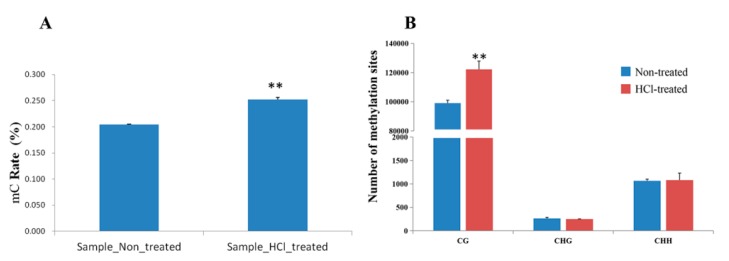
Analysis of methylation levels and methylation sites of different samples. Percentage methylation in different samples (**A**) and the numbers of different methylation sites (**B**). Three biological replicates comprised each sample type. Significant differences are indicated by ** *p* < 0.01.

**Figure 4 ijms-21-00671-f004:**
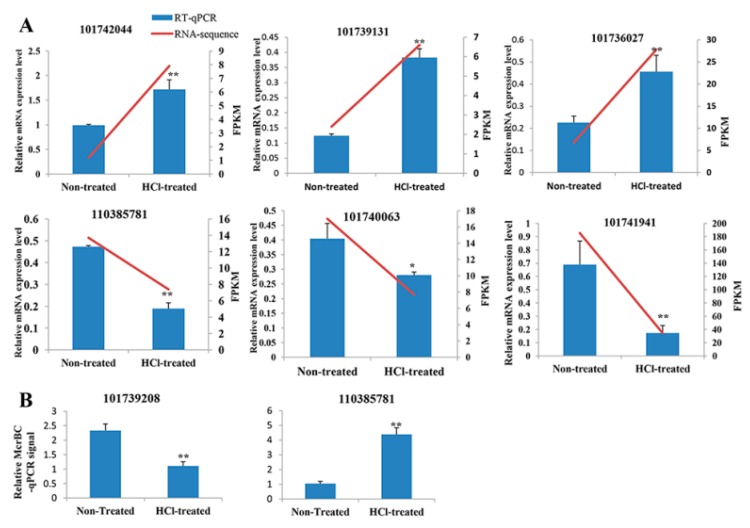
Validation of RNA-seq and whole-genome sequencing coupled with bisulfite DNA treatment (WGBS). Three genes that were significantly upregulated (101742044, 101739131 and 101739208) and three genes that were significantly downregulated (110385781, 101740063 and 101741941) from RNA-seq analyses were selected for mRNA quantification using RT-qPCR (**A**). DNA methylation levels of the hypermethylated gene, (101739208) and the hypomethylated gene, (110385781) from our WGBS were analyzed using methylation-sensitive McrBC-qPCR (**B**). Methylated DNA was digested by McrBC; thus, higher qPCR signals indicate lower methylation levels, and lower qPCR signals indicate higher methylation levels. Significant differences are indicated by ** *p* < 0.01 and * *p* < 0.05.

**Figure 5 ijms-21-00671-f005:**
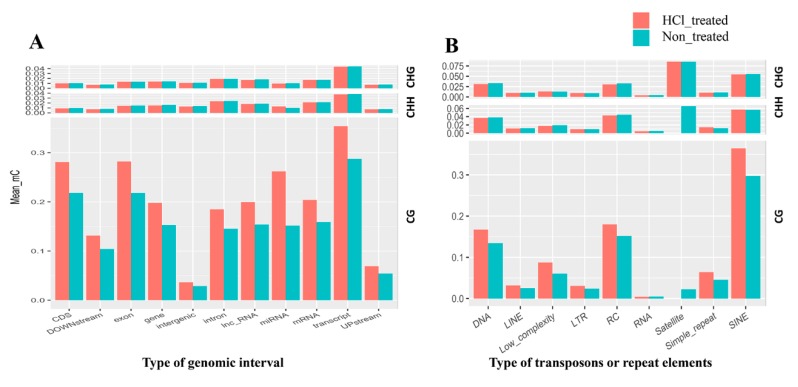
DNA methylation patterns in different genomic regions. Methylation percentages of different genetic elements, including CDS, downstream, exon, gene, intergenic, intron, inc-RNA, miRNA, mRNA, transcript, and upstream (**A**). Methylation percentages of transposon elements (TEs) and repeat elements (REs) (**B**). DNA, long interspersed nuclear elements (LINE), short interspersed nuclear elements (SINE), retrotransposons (RC) and long terminal repeats (LTR) are classified as TE, and Simple_repeat, Low_complexity, satellite and RNA are classified as RE.

**Figure 6 ijms-21-00671-f006:**
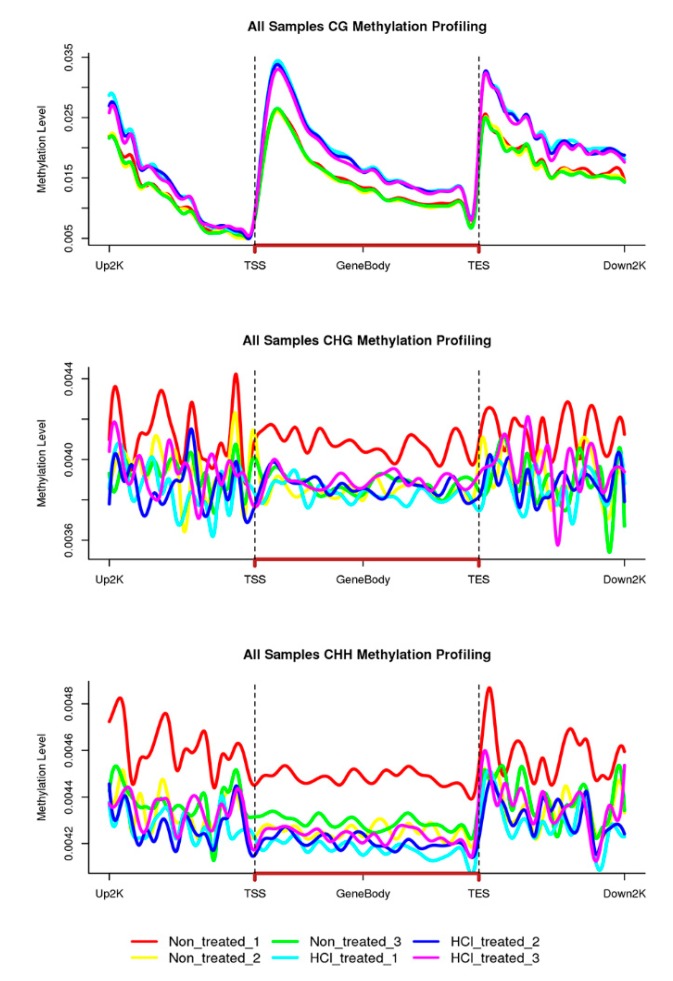
Percentage methylation levels in gene bodies and 2 kb upstream and downstream regions. Different coloured lines represent different experimental repetitions.

**Figure 7 ijms-21-00671-f007:**
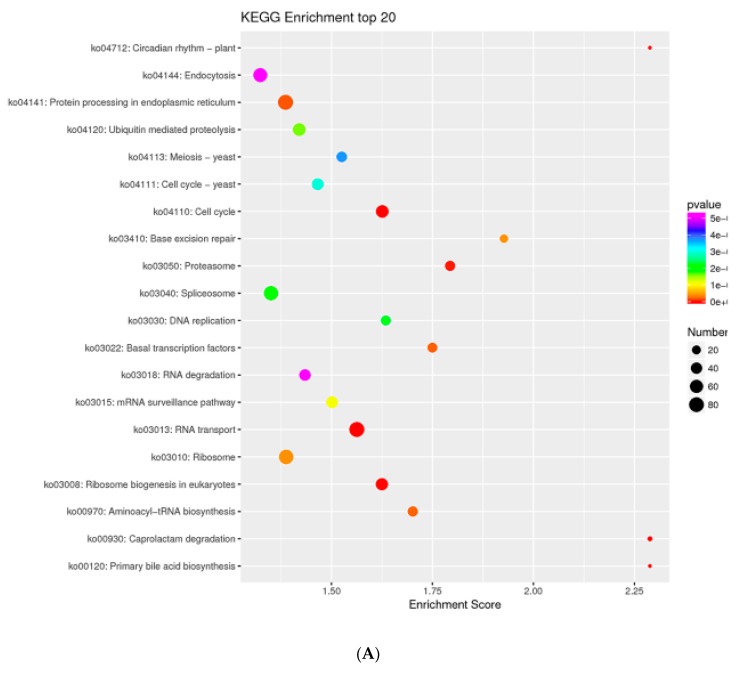
KEGG pathway enrichment of different methylation genes (DMGs) and differentially methylated promoters (DMPs) following HCl treatment. KEGG pathway enrichment of DMGs located in differentially methylated regions (DMRs) (**A**). DMPs located in DMRs (**B**). The circle size represents gene numbers, and the colour represents the Q value.

**Figure 8 ijms-21-00671-f008:**
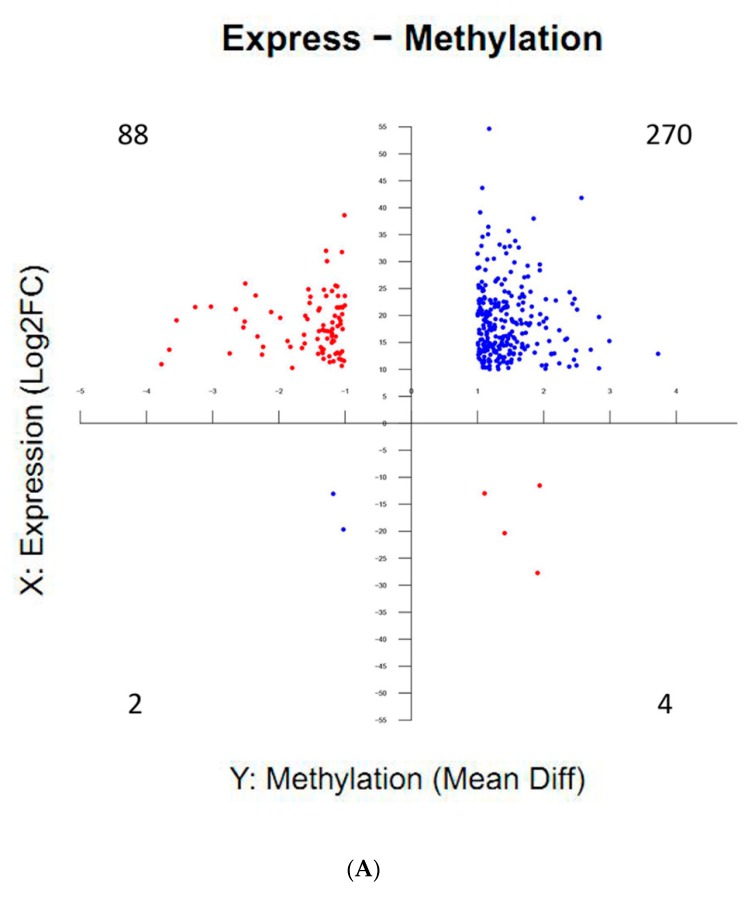
Distribution and functional enrichment analysis of significant differences in methylation and expression levels in methylome- and transcriptome-associated genes (MTGs). The distribution of MTGs in four different quadrants; the number in each quadrant refers to the number of MTGs in that quadrant (**A**). MTGs were classified into cellular components, molecular function and biological processes by WEGO according to GO terms (**B**). KEGG pathway enrichment of MTGs. The circle size represents gene numbers, and the colour represents the Q value (**C**).

**Figure 9 ijms-21-00671-f009:**
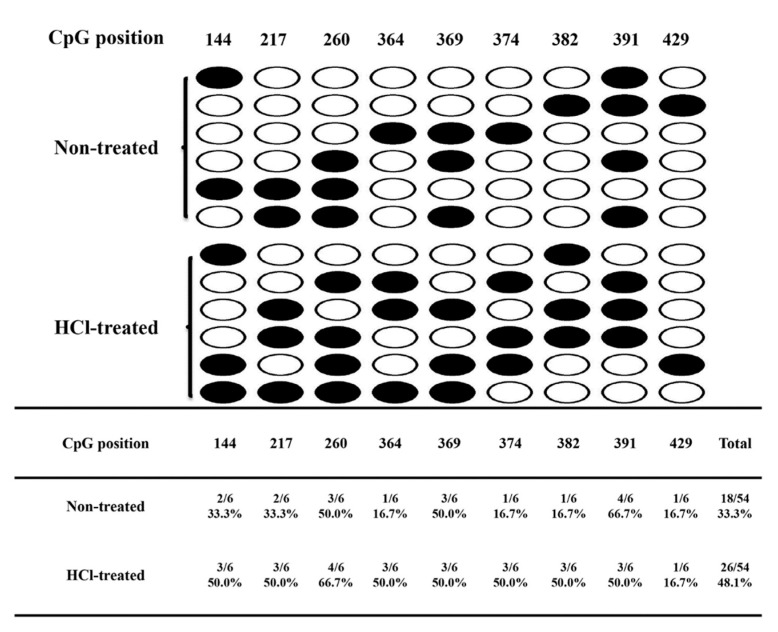
Bisulfite sequencing validation of the hypermethylated gene, *G2E3* (101739208).The number indicates the CG position of the target gene, and the percentage represents the statistical mean methylation level of the CG site or the whole CG site. Dark circles indicate CG site methylation, and open circles refer to unmethylated cytosines. Each row consists of a single sequenced clone.

**Table 1 ijms-21-00671-t001:** Statistical analysis of RNA-seq data obtained of diapause-destined eggs (non-treated) and diapause-terminated eggs (HCl-treated).

SampleName	Raw Reads	Clean Reads	Clean Bases	Total Mapped	≥Q30	GC Content
HCl-treated_1	48,961,256	47,914,231	7,012,354,124	95.49%	95.17%	43.33%
HCl-treated_2	49,151,346	48,032,314	7,032,354,785	95.33%	95.02%	43.62%
HCl-treated_3	49,564,521	48,462,345	7,094,523,651	95.41%	95.00%	43.46%
Non-treated_1	49,254,256	48,108,659	7,045,898,541	95.33%	94.88%	43.63%
Non-treated_2	49,422,345	48,188,754	7,034,562,547	94.78%	94.84%	43.77%
Non-treated_3	49,802,145	48,657,854	7,117,854,216	95.11%	95.03%	43.66%
HCl-treated vs. Non-treated			**DEGs**	**Upregulated**		**Downregulated**
Quantity			1732	856		876

**Table 2 ijms-21-00671-t002:** Summary ofWGBSdataof diapause-destined eggs (Non-treated) and diapause-terminated eggs (HCl-treated).

Samples Name	Clean Reads	Clean Base	Clean Reads Percentage	Clean Base Percentage	GC Content	>Q30
Non-treated_1	93,793,430	13,090,963,533	98.79%	98.49%	19.29%	94.81%
Non-treated_2	96,273,278	13,450,975,096	98.08%	97.88%	19.27%	92.81%
Non-treated_3	92,904,772	12,983,974,042	98.08%	97.91%	19.22%	92.82%
HCl-treated_1	98,947,490	13,832,834,274	98.03%	97.89%	19.18%	92.85%
HCl-treated_2	95,316,260	13,313,691,168	98.38%	98.15%	19.26%	93.48%
HCl-treated_3	89,068,596	12,445,567,794	98.12%	97.93%	19.26%	92.95%

**Table 3 ijms-21-00671-t003:** Methylomes- and transcriptome-associated genes involved in metabolism biosynthesis.

GeneID	Synonym	Methylation Difference Level (%)	Methylation *p*-Value	Expression Foldchange	Expression *p*-Value	Description
N-glycan biosynthesis						
LOC101745965	FucT6	15.19663	2.85 × 10^−25^	4.233024	8.12 × 10^−2^^3^	alpha-(1,6)-fucosyltransferase-like [*Bombyx mori* (domestic silkworm)]
LOC101739008	ALG9	17.21088	1.25 × 10^−25^	2.306166	6.43 × 10^−2^^8^	alpha-1,2-mannosyltransferase ALG9 [*Bombyx mori* (domestic silkworm)]
LOC101742247	Stt3b	22.27647	2.30 × 10^−^^10^	2.125494	3.94 × 10^−2^^8^	dolichyl-diphosphooligosaccharide--protein glycosyltransferase subunit STT3B [*Bombyx mori* (domestic silkworm)]
LOC101744335	alpha-Man-Ia	12.0725	3.80 × 10^−^^7^	2.024585	1.34 × 10^−^^17^	mannosyl-oligosaccharide alpha-1,2-mannosidase IA [*Bombyx mori* (domestic silkworm)]
LOC101742043	GmII	10.32376	5.80 × 10^−^^16^	2.1135	2.50 × 10^−^^65^	alpha-mannosidase 2 [*Bombyx mori* (domestic silkworm)]
Glycosaminoglycan biosynthesis—heparan sulfate/heparin						
LOC101741346	EXT2	22.1426	1.38 × 10^−^^16^	2.396993	7.06 × 10^−^^6^	exostosin-2 [*Bombyx mori* (domestic silkworm)]
LOC101736578	Hs2st	21.04841	3.40 × 10^−^^15^	2.298892	2.05 × 10^−^^14^	heparin sulfate O-sulfotransferase [*Bombyx mori* (domestic silkworm)]
LOC101737390	HS6ST2	21.02652	1.21 × 10^−^^8^	3.199666	4.66 × 10^−^^7^	heparan-sulfate 6-O-sulfotransferase 2 [*Bombyx mori* (domestic silkworm)]
Glutathione metabolism						
LOC101745413	Oplah	14.92435	2.49 × 10^−^^6^	2.405576	3.76 × 10^−^^14^	5-oxoprolinase [*Bombyx mori* (domestic silkworm)]
LOC101742094	Srm	10.35505	1.84 × 10^−5^	2.181386	3.31 × 10^−^^11^	spermidine synthase [*Bombyx mori* (domestic silkworm)]
LOC692521	GSTd2	11.33547	5.47 × 10^−^^5^	0.426264	1.12 × 10^−^^13^	glutathione S-transferase delta 2[*Bombyx mori* (domestic silkworm)]
LOC100862776	Ggt1	12.9659	2.17 × 10^−^^7^	0.149641	9.93 × 10^−^^42^	gamma-glutamyl transpeptidase [*Bombyx mori* (domestic silkworm)]
Regulation of lipolysis in adipocytes						
LOC101737606	Ac76E	10.18663	5.91 × 10^−^^5^	7.139607	1.27 × 10^−^^6^	adenylate cyclase type 2 [*Bombyx mori* (domestic silkworm)]
LOC101745072	irs1-b	23.61161	1.20 × 10^−^^12^	2.851851	2.82 × 10^−^^20^	insulin receptor substrate 1-B [*Bombyx mori* (domestic silkworm)]
LOC100158253	Pi3k60	20.1611	8.42 × 10^−^^13^	2.09716	7.71 × 10^−^^44^	phosphatidylinositol 3-kinase 60 [*Bombyx mori* (domestic silkworm)]
Biosynthesis of amino acids						
LOC101740956	ACO1	25.61842	7.63 × 10^−^^8^	2.171117	6.59 × 10^−^^24^	cytoplasmic aconitate hydratase [*Bombyx mori* (domestic silkworm)]
LOC101740405	Gs2	17.49432	3.22 × 10^−^^12^	0.438415	4.33 × 10^−^^29^	glutamine synthetase 2 cytoplasmic [*Bombyx mori* (domestic silkworm)]
LOC101743550	Ald	24.49332	1.70 × 10^−^^14^	0.436375	5.81 × 10^−^^37^	fructose-bisphosphate aldolase [*Bombyx mori* (domestic silkworm)]
LOC101738188	argF	20.62089	5.46 × 10^−^^9^	0.230464	1.88 × 10^−^^79^	ornithine carbamoyltransferase-like [*Bombyx mori* (domestic silkworm)]
LOC101738186	ACO2	17.0617	6.27 × 10^−^^14^	0.415694	0	probable aconitate hydratase, mitochondrial [*Bombyx mori* (domestic silkworm)]
Glyoxylate and dicarboxylate metabolism						
LOC101740956	ACO1	25.61842	7.63 × 10^−^^8^	2.171117	6.59 × 10^−^^24^	cytoplasmic aconitate hydratase [Bombyx mori (domestic silkworm)]
LOC101740405	Gs2	17.49432	3.22 × 10^−^^12^	0.438415	4.33 × 10^−^^29^	glutamine synthetase 2 cytoplasmic [*Bombyx mori* (domestic silkworm)]
LOC101738186	ACO2	17.0617	6.27 × 10^−^^14^	0.415694	0	probable aconitate hydratase, mitochondrial [*Bombyx mori* (domestic silkworm)]

**Table 4 ijms-21-00671-t004:** Methylomes- and transcriptome-associated genes involved in phosphorylation.

GeneID	Synonym	Methylation Difference Level (%)	Methylation *p*-Value	Expression Foldchange	Expression *p*-Value	Description
Phosphatidylinositol-signaling system						
LOC101740091	Inpp5e	29.18555	0.000496	3.378116	6.58 × 10^−^^6^	72 kDa inositol polyphosphate 5-phosphatase [*Bombyx mori* (domestic silkworm)]
LOC101740803	Inpp5j	22.42952	1.11 × 10^−^^6^	2.701747	6.94 × 10^−^^9^	inositol polyphosphate 5-phosphatase K [*Bombyx mori* (domestic silkworm)]
LOC101745952	PTEN	14.48683	7.92 × 10^−^^6^	2.540607	7.55 × 10^−^^75^	phosphatidylinositol 3,4,5-trisphosphate 3-phosphatase and dual-specificity protein phosphatase PTEN [*Bombyx mori* (domestic silkworm)]
LOC101735646	Plce1	16.95742	7.17 × 10^−^^28^	2.1977	7.13 × 10^−^^10^	1-phosphatidylinositol 4,5-bisphosphate phosphodiesterase epsilon-1 [*Bombyx mori* (domestic silkworm)]
LOC100158253	Pi3k60	20.1611	8.42 × 10^−^^13^	2.09716	7.71 × 10^−^^44^	phosphatidylinositol 3-kinase 60 [*Bombyx mori* (domestic silkworm)]
LOC101743221	Inpp5a	22.18242	3.06 × 10^−^^19^	2.035092	7.69 × 10^−^^8^	type I inositol 1,4,5-trisphosphate 5-phosphatase [*Bombyx mori* (domestic silkworm)]

**Table 5 ijms-21-00671-t005:** Methylomes- and transcriptome-associated genes involved in cell cycle and apoptosis.

GeneID	Synonym	Methylation Difference Level (%)	Methylation *p*-Value	Expression Foldchange	Expression *p*-Value	Description
G1/S transition of mitotic cell cycle						
LOC101746803	Phf8	22.76216	1.34 × 10^−^^9^	4.539773	9.05 × 10^−^^12^	histone lysine demethylase PHF8 [*Bombyx mori* (domestic silkworm)]
LOC 100216493	Cyce	16.97983	6.34 × 10^−^^6^	2.868805	1.35 × 10^−^^97^	cyclin E [*Bombyx mori* (domestic silkworm)]
LOC101741790	SKP2	26.22085	0.001772	2.090556	9.75 × 10^−^^24^	S-phase kinase-associated protein 2 [*Bombyx mori* (domestic silkworm)]
LOC101738704	ACVR1	13.23511	1.51 × 10^−^^5^	2.028722	5.10 × 10^−^^59^	activin receptor type-1 [*Bombyx mori* (domestic silkworm)]
Regulation of apoptotic process						
LOC101742044	CG14226	10.69776	1.84 × 10^−^^11^	5.6281	4.06 × 10^−^^93^	cytokine receptor [*Bombyx mori* (domestic silkworm)]
LOC101739208	G2e3	16.07484	2.73 × 10^−^^14^	2.619197	4.72 × 10^−^^51^	G2/M phase-specific E3 ubiquitin-protein ligase [*Bombyx mori* (domestic silkworm)]
LOC101745221	DLG5	11.60066	1.46 × 10^−^^06^	2.141483	6.91 × 10^−^^51^	disks large homolog 5 [Bombyx mori (domestic silkworm)]
LOC101741790	SKP2	26.22085	0.001772	2.090556	9.75 × 10^−^^24^	S-phase kinase-associated protein 2 [*Bombyx mori* (domestic silkworm)]
LOC101740336	Sh3kbp1	22.78633	0.001078	2.027284	2.31 × 10^−^^45^	CD2-associated protein [*Bombyx mori* (domestic silkworm)]
LOC101738660	Ppid	21.54647	3.27 × 10^−^^19^	0.480788	6.92 × 10^−^^53^	peptidyl-prolyl cis-trans isomerase D [*Bombyx mori* (domestic silkworm)]
Signaling pathways regulating pluripotency of stem cells						
LOC101742798	Mad	12.83938	0.001686	4.322594	2.95 × 10^−^^21^	protein mothers against dpp [*Bombyx mori* (domestic silkworm)]
LOC101744534	BMPR2	23.0032	1.24 × 10^−^^6^	4.090361	2.39 × 10^−^^30^	probable serine/threonine-protein kinase DDB_G0278901 [*Bombyx mori* (domestic silkworm)]
LOC101740154	RIF1	12.9223	2.47 × 10^−^^9^	2.924023	4.05 × 10^−^^19^	telomere-associated protein RIF1 [*Bombyx mori* (domestic silkworm)]
LOC100158253	Pi3k60	20.1611	8.42 × 10^−^^13^	2.09716	7.71 × 10^−^^44^	phosphatidylinositol 3-kinase 60 [*Bombyx mori* (domestic silkworm)]
LOC101735864	KIAA0445	15.4856	8.89 × 10^−^^7^	2.064435	2.90 × 10^−^^47^	SWI/SNF-related matrix-associated actin-dependent regulator of chromatin subfamily A containing DEAD/H box 1 homolog [*Bombyx mori* (domestic silkworm)]
LOC101738704	ACVR1	13.23511	1.51 × 10^−^^5^	2.028722	5.10 × 10^−^^59^	activin receptor type-1 [*Bombyx mori* (domestic silkworm)]

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
