# Peer review of "DNA Methylation Is Correlated with Gene Expression during Diapause Termination of Early Embryonic Development in the Silkworm (Bombyx mori)"

_ijms, 2020, doi:10.3390/ijms21020671_

Round 1

Reviewer 1 Report

In the silkworm, relationship between DNA methylomes and transcriptomes in diapause-destined eggs, compared to diapause terminated eggs was analyzed.  364 genes showed different methylation and expression levels between diapause-destined and diapause terminated eggs.  This research will provide valuable information to future researches about roles of DNA methylation on gene regulation in insects.  I request the authors further improvement of the manuscript.

DNA methylation in insects still get entangled, especially in the silkworm.  The silkworm genome lacks Dnmt3 gene which can methylate DNA de novo, while Dnmt1 methylate only hemimethylate sites.  Therefore, how DNA methylation regulate gene expression in the silkworm is enigma.  To the readers not familiar in DNA methylation, authors should explain fully the function of Dnmts and loss of Dnmt3 in the silkworm genome in Introduction and Discussion sections, and describe their thought on possible mechanism of DNA methylation in the silkworm. On diapause termination, 270 hypermethylated genes were positively correlated with expression changes and 88 hypermethylated genes were negatively correlated (Fig. 8A).  For these genes, show the methylation profiles separately, as shown in Fig. 6.  Is there any difference between positively and negatively correlated genes?  If there is no difference, it should be also described.  It will provide further information on gene regulation by DNA methylation. DNA methylation was sparse in silkworm genome.  How are the methylation sites selected?  In this report, correlation with DNA methylomes and transcriptomes on the diapause termination were shown.  Does DNA methylation regulate gene transcription or gene transcription regulates DNA methylation sites?  Please discuss it or comment.

Author Response

Thank you very much for giving us an opportunity to revise our manuscript entitled “DNA methylation is correlated with gene expression during diapause termination of early embryonic development in the silkworm (Bombyx mori)” (ijms-645725), we have revised our manuscript according to reviewer’s suggestions. Our changes are highlighted in colored marker in the revised manuscript. Point-by-point replies are listed as follows:

To the readers not familiar in DNA methylation, authors should explain fully the function of Dnmts and loss of Dnmt3 in the silkworm genome in Introduction and Discussion sections, and describe their thought on possible mechanism of DNA methylation in the silkworm.

Reply: Thank you for your advice, we have revised our manuscript according to reviewer’s suggestions, and the functions of methyltransferase in silkworm were introduced. We have revised the content at Line: 46-48, 76-77.

DNA methyltransferases (DNMTs) is determinate enzymes associated with 5mC (Jurkowska et al., 2011), and are thought to be a family, primarily Dnmt1, Dnmt2 and Dnmt3 (Goll and Bestor, 2005; Okano et al., 1999).

Silkworm have only Dnmt1 and Dnmt2 but not Dnmt3,  and BmDnmt1 retained the function as maintenance DNMT, but its sensitivity to metal ions is different from mammalian Dnmt1(Mitsudome et al., 2015).

In Fig. 6. is there any difference between positively and negatively correlated genes?

Reply: Thanks for your comments, Figure 6 analyzed the average level of gene modification upstream or other regions in the whole methylated sequencing sample, which represents a trend. At Fig. 6 , the methylation modification pattern of a single gene cannot be showed.

Does DNA methylation regulate gene transcription or gene transcription regulates DNA methylation sites?

Reply: Thanks for your question, and this is a very central question for our research. In the discussion section of the article, we mentioned that DNA methylation generally functions as a repressive transcriptional signal, but it is also well known to activate gene expression. Through the methylation modification regularity and the related analysis of WGBS and RNA-seq, we speculate that CG methylation occurs in gene body regions, enhancing gene transcription for embryonic development. Therefore, for this question, we believe that methylation regulates gene transcription.

Reviewer 2 Report

The authors submitted a paper with the title “diapause termination of early embryonic development in the silkworm (Bombyx mori)”. This paper presents experimental data showing DNA methylation patterns using WGBS and RNA sequencing technologies in diapause terminated eggs, compared to diapause-destined eggs in B. mori. Their results indicate DNA methylation is essential during diapause and embryogenesis of B. mori.

specific comments.

1. The title need more accurate.

2. All figures need more clear. There are some elementary mistakes in fig.1 and fig.8

3. Authors select 3-day eggs for RNA-Seq and WGBS. Normally, B. mori eggs can keep adout 10 days. why authors choose this time point to measure?

4. In section 3.3, why authors choose the 6 genes to validate using qPCR? Because their highest expression fold change compare to control in RNA-seq? Before the validation using qPCR, authors can show a list of differentially expressed genes.

Author Response

Thank you very much for giving us an opportunity to revise our manuscript entitled “DNA methylation is correlated with gene expression during diapause termination of early embryonic development in the silkworm (Bombyx mori)” (ijms-645725), we have revised our manuscript according to reviewer’s suggestions. Our changes are highlighted in colored marker in the revised manuscript. Point-by-point replies are listed as follows:

The title need more accurate.

Reply: Our manuscript entitled is “DNA methylation is correlated with gene expression during diapause termination of early embryonic development in the silkworm (Bombyx mori)”, but the latest edition of the manuscript for revision is “diapause termination of early embryonic development in the silkworm (Bombyx mori)”. There must be a mistake for the peer review edition.

All figures need more clear. There are some elementary mistakes in fig.1 and fig.8

Reply: We are sorry for the unclear figure in previous manuscript, and will reupload the clearer figure. The errors in figure 1 and figure 8 are edited to ensure accuracy. There are some elementary mistakes in fig.1 and fig.8 in peer review edition, and it may be caused by the software, There is no mistakes with the original picture we uploaded separately, we ask the editor help to solve it.

Authors select 3-day eggs for RNA-Seq and WGBS. Normally, B. mori eggs can keep adout 10 days. why authors choose this time point to measure?

Reply: Diapause-destined eggs, eggs completely enter diapause at 3-day after oviposition with non-HCl treated (control group). If diapause-destined eggs were treated with 1.075 g/L HCl at 46°C for 5 min, 24 h after oviposition, the eggs will terminat diapause at 48 h after  HCl treatment (3-day after oviposition, experimental group). The eggs stage of  the experimental group and control group were consistent. Hence, we select 3-day eggs for RNA-Seq and WGBS to analyze the effect of methylation on diapause of silkworm. We have revised the content at Line: 82-85, and 110-116.

In section 3.3, why authors choose the 6 genes to validate using qPCR? Because their highest expression fold change compare to control in RNA-seq? Before the validation using qPCR, authors can show a list of differentially expressed genes.

Reply: GO and KEGG pathway analysis identified some important pathways that may be related to diapause, then we selected genes from these pathways for verification based on their functions. The genes we selected all appear in the table of pathway gene analysis. a list of differentially expressed genes was presented in Supplementary Table S2.

Reviewer 3 Report

In this study Li et al investigated the methylation and gene expression in the early embryonic development of the silkworm. They compared the gene expression and methylation between the diapause-destined and diapause terminated eggs and found a number of genes with differential expression as well as the differential methylation sites. They identified pathways that could possibly be concerned with the embryonic development. The methylation regulation is of significance in a wide variety of biological process and thus this study can be perceived as an essential milestone to clarify the mechanism of the silkworm embryonic development. The followings are questions and comments that should be revised so that it can be published.

<Major points>

I propose the authors should deposit the raw NGS data on the public database (NCBI, DDBJ or others).

Line 164, “diapause-destined eggs and diapause terminated eggs”. The authors used samples collected from “48 h after HCl treatment” for the former and “48 h after without HCl treatment” for the latter. Is there any difference of the embryonic stage for these samples? Show the detailed developmental stage for these samples.

I propose the Figure 9 should be transferred to the Supplementary Materials.

<Minor points>

Line 53. I guess WGBS means “whole-genome bisulfite sequencing”, not “whole-genome sequencing.

Line 64. Lepidoptera -> lepidopteran

Line 95. five females -> five broods?

Line 96. however eggs were -> in which the eggs were

Line 123. Did the authors really use Hiseq X, not Hiseq4000? I guess the Illumina permits Hiseq X application just for the whole genome analysis, not for the transcriptomic analysis.

Line 165. CG content -> GC content.

Line 169. Supplementary Table S2 -> Supplementary Table S2.

Line 170. catalytic activity. -> catalytic activity

Table 1. Non_treated_2 -> Non-treated_2

Table 1. For the raw reads, clean reads and clean bases, show the detailed number (not xxM or xxG), like as in Table 2.

Line 197 and 198. Show the explanation of “HCl-treated” and “non HCl-treated” in the line 164, not here.

Line 220. with RNA-Seq -> with RNA-Seq results

Line 224. increased -> decreased?

Line 226. decreased -> increased?

Line 248. (long terminal repeats) LTRs -> long terminal repeats (LTRs)

Line 254. Downstream -> downstream

Line 261. regularity -> bias?

Line 268. What is the definition of “promoters”? 2 kb upstream regions?

Line 274. As shown (Figure 7A), -> As shown in Figure 7A,

Line 316-317. During the embryonic developmental stage of diapause eggs terminate the diapause, energy metabolism and material synthesis begin to play important roles. -> What does this sentence mean?

Line 323. pathway in which -> pathway, in which

Line 324. are enhancement -> are enhanced

Line 332. A previous studies -> A previous study

Line 345. G1, once diapause terminates… -> G1. Once diapause terminates…

Line 346. M phase, cell division -> M phase, and cell division

Line 352. ligase (G2E3) was upregulated -> ligase (G2E3), was upregulated

Line 353. which occurs every two days -> which occurs two days

Line 356. increased gene expression, those MTGs can -> increased gene expression. Those MTGs can

Line 373. shown that ~0.21%-0.26% -> shown ~0.21%-0.26%

Line 376. When compared to eggs, -> When comparing eggs,

Line 404. SINES -> SINEs

Line 420. silkworm embryogenesis, was followed -> silkworm embryogenesis was followed

Author Response

Thank you very much for giving us an opportunity to revise our manuscript entitled “DNA methylation is correlated with gene expression during diapause termination of early embryonic development in the silkworm (Bombyx mori)” (ijms-645725), we have revised our manuscript according to reviewer’s suggestions. Our changes are highlighted in colored marker in the revised manuscript. Point-by-point replies are listed as follows:

<Major points>

I propose the authors should deposit the raw NGS data on the public database (NCBI, DDBJ or others).

Reply: Thanks for your advice. We are uploading the raw data to the NCBI database, which may be completed in a few days due to the large amount of data.

Line 164, “diapause-destined eggs and diapause terminated eggs”. The authors used samples collected from “48 h after HCl treatment” for the former and “48 h after without HCl treatment” for the latter. Is there any difference of the embryonic stage for these samples? Show the detailed developmental stage for these samples.

Reply: In order to artificial terminated the diapause, diapause-destined eggs were treated with 1.075 g/L HCl at 46°C for 5 min at 24 h after oviposition, and then the eggs for the experimental group were collected at 48 h after HCl treatment. So the time point is at 72 h (3-day stage) after oviposition. At this stage, silkworm eggs completely terminate diapause. By contrast, for the control group, eggs completely enter diapause at 3-day stage after oviposition with non-HCl treated. The eggs stage of  the experimental group and control group were consistent. We have revised the content at Line: 82-85, and 110-116.

I propose the Figure 9 should be transferred to the Supplementary Materials.

Reply: Thank you for your thoughtful advice. The gene 101739208, one of the six MTGs, was a homologue of the G2/M phase-specific E3 ubiquitin-protein ligase (G2E3), Previous research has shown that G2E3 is essential for early embryonic development in preventing apoptotic death, and it strengthens the synthesis and transport of genetic material and energy in the M phase, and the development of diapause destined embryos is arrested during the G2 cell cycle stage, immediately after mesoderm formation. Therefore, we speculate that G2E3 may be related to diapause of silkworm embryos and may function through methylation. The analysis of specific methyl sites of G2E3 in diapause-destined eggs and diapause terminated eggs will help to further clarify the mechanism of  regulation of diapause. Hence, the content in figure 9 is critical to the follow-up to this article, and we request that it be left in the body of article.

<Minor points>

Line 53. I guess WGBS means “whole-genome bisulfite sequencing”, not “whole-genome sequencing”.

Reply: Thank you for providing the professional suggestions. The sentence has been revised to “whole-genome sequencing, coupled with bisulfite DNA treatment (WGBS)”.

Line 64. Lepidoptera -> lepidopteran

Reply: Thank you for providing the professional suggestions. The word has been revised to “lepidopteran”

Line 95. five females -> five broods?

Reply: Our material was obtained by sampling a mixture of five mothers laying eggs, we think “five females” is an accurate description.

Line 96. however eggs were -> in which the eggs were

Reply: Thank you for providing the professional suggestions. The sentence has been revised to “in which the eggs were”.

Line 123. Did the authors really use Hiseq X, not Hiseq4000? I guess the Illumina permits Hiseq X application just for the whole genome analysis, not for the transcriptomic analysis.

Reply: We confirmed with the sequencing company that we were actually using Illumina HiSeq X Ten for transcriptome sequencing.

Line 165. CG content -> GC content.

Reply: Thank you for providing the professional suggestions. The sentence has been revised to “GC content”

Line 169. Supplementary Table S2 -> Supplementary Table S2.

Reply: Thank you for providing the professional suggestions, and The excess punctuation has been removed.

Line 170. catalytic activity. -> catalytic activity

Reply: Thank you for providing the professional suggestions, and The excess punctuation has been removed.

Table 1. Non_treated_2 -> Non-treated_2

Reply: Thank you for providing the professional suggestions. The sentence has been revised to “Non-treated_2”

Table 1. For the raw reads, clean reads and clean bases, show the detailed number (not xxM or xxG), like as in Table 2.

Reply: Thank you for providing the professional suggestions. we have showed the detailed number of the raw reads, clean reads and clean bases in our revised article.

Line 197 and 198. Show the explanation of “HCl-treated” and “non HCl-treated” in the line 164, not here.

Reply: Thank you for providing the professional suggestions. We have showed the explanation of “HCl-treated” and “non HCl-treated” in the line 164.

Line 220. with RNA-Seq -> with RNA-Seq results

Reply: Thank you for providing the professional suggestions. The sentence has been revised to “with RNA-Seq results”.

Line 224. increased -> decreased?

Reply: We have confirmed that there are no errors here. Methylated DNA was digested by McrBC, thus higher qPCR signals indicate lower methylation levels and lower qPCR signals indicate higher methylation levels. We have described the related contents in detail in the figure legend of Figure 4.

Line 226. decreased -> increased?

Reply: We have confirmed that there are no errors here. Methylated DNA was digested by McrBC, thus higher qPCR signals indicate lower methylation levels and lower qPCR signals indicate higher methylation levels. We have described the related contents in detail in the figure legend of Figure 4.

Line 248. (long terminal repeats) LTRs -> long terminal repeats (LTRs)

Reply: Thank you for providing the professional suggestions. The sentence has been revised to “long terminal repeats (LTRs)”.

Line 254. Downstream -> downstream

Reply: Thank you for providing the professional suggestions. The word has been revised to “downstream”.

Line 261. regularity -> bias?

Reply: Thank you for providing the professional suggestions. Here we just describe the consistent regularity of the changes, without statistical analysis.

Line 268. What is the definition of “promoters”? 2 kb upstream regions?

Reply: The analysis of the promoter is not designed here.

Line 274. As shown (Figure 7A), -> As shown in Figure 7A,

Reply: Thank you for providing the professional suggestions. The sentence has been revised to “As shown in Figure 7A”.

Line 316-317. During the embryonic developmental stage of diapause eggs terminate the diapause, energy metabolism and material synthesis begin to play important roles. -> What does this sentence mean?

Reply: Thank you for providing the professional suggestions. The sentence has been revised to “In embryonic development, when the diapause eggs have terminated the diapause, energy metabolism and material synthesis begin to be activated”.

Line 323. pathway in which -> pathway, in which

Reply: Thank you for providing the professional suggestions. The sentence has been revised to “pathway, in which”.

Line 324. are enhancement -> are enhanced

Reply: Thank you for providing the professional suggestions. The sentence has been revised to “are enhanced”.

Line 332. A previous studies -> A previous study

Revise: Thank you for providing the professional suggestions. The sentence has been revised to “A previous study”.

Line 345. G1, once diapause terminates… -> G1. Once diapause terminates…

Reply: Thank you for providing the professional suggestions. The sentence has been revised to “G1. Once diapause terminates…”.

Line 346. M phase, cell division -> M phase, and cell division

Reply: Thank you for providing the professional suggestions. The sentence has been revised to “M phase, and cell division”.

Line 352. ligase (G2E3) was upregulated -> ligase (G2E3), was upregulated

Reply: Thank you for providing the professional suggestions. The sentence has been revised to “ligase (G2E3), was upregulated”.

Line 353. which occurs every two days -> which occurs two days

Reply: Thank you for providing the professional suggestions. The sentence has been revised to “which occurs two days”.

Line 356. increased gene expression, those MTGs can -> increased gene expression. Those MTGs can

Reply: Thank you for providing the professional suggestions. The sentence has been revised to “increased gene expression. Those MTGs can”.

Line 373. shown that ~0.21%-0.26% -> shown ~0.21%-0.26%

Reply: Thank you for providing the professional suggestions. The sentence has been revised to “shown ~0.21%-0.26%”.

Line 376. When compared to eggs, -> When comparing eggs,

Reply: Thank you for providing the professional suggestions. The sentence has been revised to “When comparing eggs”.

Line 404. SINES -> SINEs

Reply: Thank you for providing the professional suggestions. The word has been revised to “SINEs”.

Line 420. silkworm embryogenesis, was followed -> silkworm embryogenesis was followed

Reply: Thank you for providing the professional suggestions. The word has been revised to “silkworm embryogenesis was followed”.